# Comparative Analyses of 35 Complete Chloroplast Genomes from the Genus *Dalbergia* (Fabaceae) and the Identification of DNA Barcodes for Tracking Illegal Logging and Counterfeit Rosewood

Zhou Hong [1] , Wenchuang He [2] , Xiaojing Liu [1], Luke R. Tembrock [3], Zhiqiang Wu [2,4,*] , Daping Xu [1,*] and Xuezhu Liao [2,*]

1 State Key Laboratory of Tree Genetics and Breeding, Research Institute of Tropical Forestry, Chinese Academy of Forestry, Guangzhou 510520, China; hzhou1981@sina.com (Z.H.); xjlliucaf@163.com (X.L.)
2 Shenzhen Branch, Guangdong Laboratory for Lingnan Modern Agriculture, Genome Analysis Laboratory of the Ministry of Agriculture, Agricultural Genomics Institute at Shenzhen, Chinese Academy of Agricultural Sciences, Shenzhen 518120, China; hewenchuang@caas.cn
3 Department of Agricultural Biology, Colorado State University, Fort Collins, CO 80523, USA; luke.tembrock@colostate.edu
4 Kunpeng Institute of Modern Agriculture at Foshan, Foshan 528200, China
* Correspondence: wuzhiqiang@caas.cn (Z.W.); gzfsrd@163.com (D.X.); liaoxuezhu@caas.cn (X.L.)

**Abstract:** The genus *Dalbergia* contains more than 200 species, several of which are trees that produce traditional medicines and extremely high-value timber commonly referred to as rosewood. Due to the rarity of these species in the wild, the high value of the timber, and a growing international illicit trade, CITES (Convention on International Trade in Endangered Species of Wild Fauna and Flora) has listed the entire genus in appendix II and the species *Dalbergia nigra* in appendix I because species in this genus are considered at risk of extinction. Given this, and the fact that species or even genus level determination is nearly impossible from cut timber morphology, alternative molecular methods are needed to identify and track intercepted rosewood. To better identify rosewood using molecular methods, we sequenced and assembled eight chloroplast genomes including *D. nigra* as well as conducted comparative analyses with all other available chloroplast genomes in *Dalbergia* and closely related lineages. From these analyses, numerous repeats including simple sequence repeats (SSR) and conserved nucleotide polymorphisms unique to subclades within the genus were detected. From phylogenetic analysis based on the CDS from 77 chloroplast genes, the groups Siam rosewood and scented rosewood resolved as monophyletic, supporting the morphological traits used to delimit these species. In addition, several instances of paraphyly and polyphyly resulting from mismatches between taxonomic determinations and phylogenetic tree topology were identified. Ultimately, the highly variable regions in the chloroplast genomes will provide useful plastid markers for further studies regarding the identification, phylogeny, and population genetics of *Dalbergia* species, including those frequently intercepted in illegal trade.

**Keywords:** forensic biology; phylogeny; CITES; endangered species; forest conservation

## 1. Introduction

The genus *Dalbergia* L.f. in the tribe Dalbergieae (DC.) Cardoso and family Fabaceae is made up of approximately 275 species of trees, shrubs, and lianas. The genus is widely distributed worldwide, with species occurring in the tropics and subtropics of South and Central America, Africa, Madagascar, and Asia [1–3]. Several tree species in the genus are highly valued for producing premium darkly colored, dense, and sometimes fragrant wood used in a variety of applications such as the production of musical instruments, traditional medicine, fine furniture, cabinetry, and veneers. By way of example, rosewood

furniture in China can range from thousands of dollars for an ornate chair to millions of dollars (USD) for a bed frame [3]. The main species used to produce high-quality timber are loosely grouped into the three categories (made up of several species in each group) black rosewood, scented rosewood, and Siam rosewood, based mainly on characteristics of the wood. The black and Siam rosewoods are the most highly valued [4,5].

The extremely high value of rosewood has led to large scale illegal logging and international trade involving numerous countries where rosewood is native. The main centers of rosewood logging are found in Central and South America [6,7], Africa and Madagascar [8,9], and Southeast Asia [10,11]. Madagascar is an area of particular concern given the high number of endemic *Dalbergia* species (42 spp.) and the presence of organized syndicates taking advantage of the political and economic instability in the country [12–14]. Rosewood is considered the most trafficked group of endangered species in the world with the value of global seizures exceeding ivory, rhino horn, and big cats combined [15].

Because of illegal trade, overexploitation, and the similarity of wood between some species, the entire *Dalbergia* genus is protected under CITES (Convention on International Trade in Endangered Species of Wild Fauna and Flora) appendix II [16]. The Brazilian species *Dalbergia nigra* (Vell.) Benth endemic to the Bahia interior forest ecoregion, is considered highly threatened with extinction and has been listed in appendix I of CITES [16] prohibiting all international trade. In addition to listing in appendix II of CITES [16], *D. fusca* Pierre and *D. odorifera* T.C. Chen are also listed in China's key protection list. Given the illegal trade in rosewood and difficulty in differentiating Dalbergieae species based on a lack of morphological differences in confiscated specimens, a means of effectively identifying different rosewood species is needed. In parallel with tracking illegal trade of rosewood, programs that could be developed to produce licit rosewood timber will also benefit from routine molecular identification and tracking as part of a robust certification program.

In studies of plant biology, many different kinds of molecular markers have been developed, including SSRs (simple sequence repeats), and a variety of short (500–1000 bp) DNA sequences referred to as barcodes for the identification of species and populations [17]. The cost of sequencing has rapidly decreased in recent years, with concurrent increases in throughput making the discovery and development of diagnostic markers more efficient and cost-effective even for a large number of species [18]. Given the utility of DNA-barcode-based identification, numerous different barcode regions from different cellular compartments have been proposed in plants [19]. Hassold et al. [20] analyzed DNA barcoding (*matK*, *rbcL*, *trnL* (UAA)) of *Dalbergia* and found that subgroup-based species identification had a higher success rate, but this method also had limitations on distinguishing individual species. In recent years, super DNA barcodes have been applied to entire chloroplast genome sequences, as in the comparative analyses of complete plant chloroplast genomes [21–23]. These studies have also been carried out in *Dalbergia* and are mainly focused on clade of scented rosewood and Siam rosewood [2]. Unlike the relatively large and complex nuclear genome, the chloroplast genome has several advantages, including uniparental inheritance, high information content (in variable sites), very low recombination rates, and high copy number, making chloroplast genomes and chloroplast barcodes ideally suited for studies in plant systematics [24,25], population genetics [26,27] and plant taxonomy [19]. Moreover, this high copy number of chloroplasts enables chloroplast DNA to be detected from wood products in which DNA amplification rates varied from parts of wood [28]. The conserved gene content and structural arrangement of chloroplast genomes in two inverted repeat regions (IRs), a long single-copy region (LSC), and a short single-copy region (SSC) make assembly and alignment of chloroplast genomes more complete and less error-prone than with plant nuclear or mitochondrial genomes. Additionally, because a large number of complete chloroplast genomes are available in public databases, comparative analyses and searches are more accurate and thorough than with any other complete genome data in plants. Based on the advantages outlined

above, the whole chloroplast genome and barcodes derived therefrom would be ideal for identifying *Dalbergia* species from wood samples.

To isolate informative molecular markers useful for species identification from wood material, we conducted the following analyses: (1) sequenced, assembled, and annotated the chloroplast genomes of eight *Dalbergia* species (including the CITES listed appendix I *D. nigra*, which has never been reported) using next-generation sequencing methods; (2) conducted a phylogenetic analysis to infer the relationships among *Dalbergia* species; and (3) comprehensively analyzed the highly variable regions and conserved nucleotide sites unique to each clade in *Dalbergia* that could be employed for DNA-barcode-based identification.

## 2. Results and Discussion

### 2.1. Complete Chloroplast Genomes

The complete chloroplast genome lengths of the eight *Dalbergia* species sequenced as part of this study range from 155,330 bp to 156,697 bp and were all typically circular in structure (Figure 1, Supplemental Figure S1, Table 1). The length of each chloroplast genome was *D. nigra* (MT644130) 155,330 bp, *D. hupeana* Hance (MT644129) 155,829 bp, *D. fusca* (MT644128) 156,033 bp, *D. odorifera* (MT644131) 156,064 bp, *D. tonkinensis* Prain (MT644133) 156,087 bp, *D. bariensis* Pierre (MT644134) 156,544 bp, *D. cochinchinensis* Pierre (MT644135) 156,576 bp, and *D. oliveri* Prain (MT644132) 156,697 bp, which were consistent with other chloroplast genomes in *Dalbergia* (Song et al. 2019). The IR regions are also similar in length, ranging from 25,469 to 25,720 bp, separated by an LSC region (85,110–86,036 bp) and an SSC region (18,856–19,427 bp; Table 1; Supplemental Table S1). In general, the genome sequences of species in *Dalbergia* were similar in length.

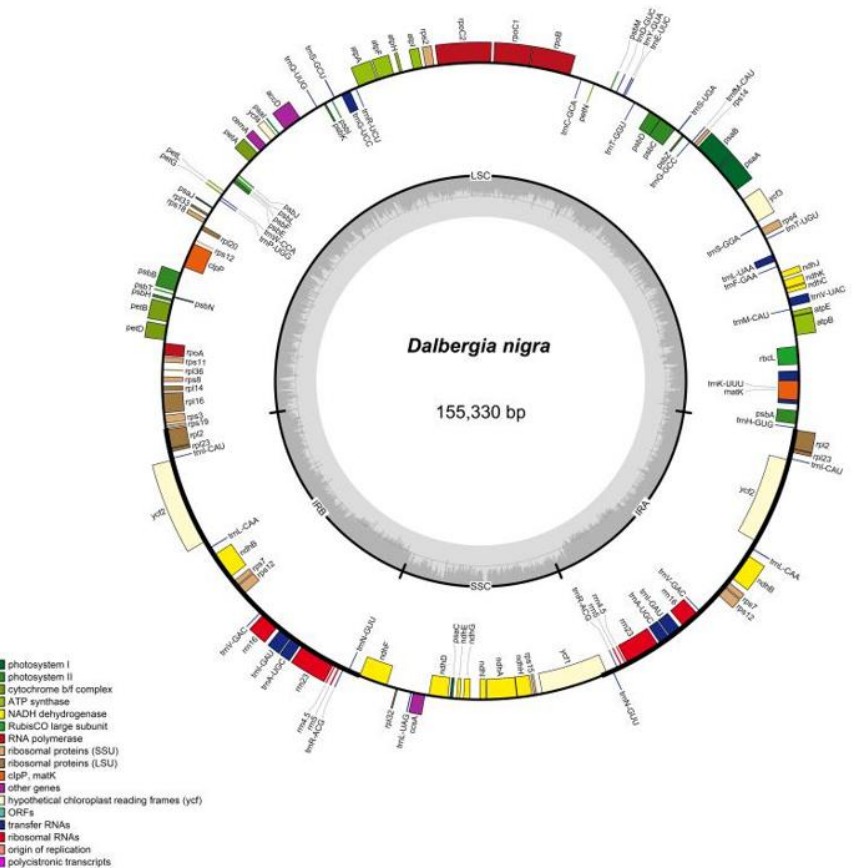

**Figure 1.** Gene map of the *D. nigra* chloroplast genome. Genes shown inside the circle are transcribed clockwise, and those outside are transcribed counterclockwise. The darker gray in the inner circle corresponds to the GC content. The IRA and IRB (two inverted repeating regions); LSC (long single-copy region); and SSC (short single-copy region) are indicated outside of the GC content.

**Table 1.** Summary of the complete chloroplast genomes sequenced for this study.

| GB ID | Species | TL | Genes | tRNA | rRNA | GC% | LSC | IRB | SS | IRA |
|-------|---------|----|-------|------|------|-----|-----|-----|-----|-----|
| MT644128 | *Dalbergia fusca* | 156,033 | 83 | 37 | 8 | 36.08 | 85,475 | 25,711 | 19,131 | 25,716 |
| MT644129 | *Dalbergia hupeana* | 155,829 | 83 | 37 | 8 | 36.19 | 85,304 | 25,680 | 19,168 | 25,677 |
| MT644130 | *Dalbergia nigra* | 155,330 | 82 | 37 | 8 | 36.05 | 85,110 | 25,469 | 19,282 | 25,469 |
| MT644131 | *Dalbergia odorifera* | 156,064 | 83 | 37 | 8 | 36.09 | 85,804 | 25,702 | 18,856 | 25,702 |
| MT644132 | *Dalbergia oliveri* | 156,697 | 82 | 37 | 8 | 35.96 | 86,036 | 25,691 | 19,278 | 25,692 |
| MT644133 | *Dalbergia tonkinensis* | 156,087 | 83 | 37 | 8 | 36.09 | 85,763 | 25,720 | 18,884 | 25,720 |
| MT644134 | *Dalbergia bariensis* | 156,544 | 83 | 37 | 8 | 35.94 | 85,765 | 25,675 | 19,427 | 25,677 |
| MT644135 | *Dalbergia cochinchinensis* | 156,576 | 83 | 37 | 8 | 36.08 | 85,886 | 25,682 | 19,326 | 25,682 |

In addition to being similar in genome length and overall structure, the number and types of genes are also very similar among the newly sequenced genomes. The annotated coding genes included 45 photosynthesis-related genes (*atpA, atpB, atpE, atpF, atpH, atpI, cemA, ndhA, ndhB, ndhC, ndhD, ndhE, ndhF, ndhG, ndhH, ndhI, ndhJ, ndhK, petA, petB, petD, petG, petL, petN, psaA, psaB, psaC, psaI, psaJ, psbA, psbB, psbC, psbD, psbE, psbF, psbH, psbI, psbJ, psbK, psbL, psbM, psbN, psbT, psbZ,* and *rbcL*), 20 ribosomal protein genes (*rpl14, rpl16, rpl2, rpl20, rpl23, rpl32, rpl33, rpl36, rps11, rps12, rps14, rps15, rps16, rps18, rps19, rps2, rps3, rps4, rps7,* and *rps8*), 4 transcription/translation genes (*rpoA, rpoB, rpoC1,* and *rpoC2*), 4 miscellaneous protein genes (*accD, ccsA, clpP* and *matK*), and 4 conserved ORFs, (*ycf1-4*) (). Introns were found in the 12 coding genes *atpF, clpP, ndhA, ndhB, petB, petD, rpl16, rpl2, rpoC1, rps12, rps16,* and *ycf3*. Among the intron-containing genes, *ycf3* and *clpP* contained two introns, while the 10 other genes contained a single intron.

Along with the 8 *Dalbergia* chloroplast genomes newly sequenced in this study, an additional 27 published genomes from *Dalbergia*, 4 from *Arachis* L., 2 from *Pterocarpus* Jacq. (which also produce a rosewood-type timber), 1 from *Tipuana tipu* (Benth.) Kuntze, and an outgroup species *Amorpha fruticosa* L. were employed where larger comparative analysis was appropriate (Supplemental Tables S1 and S3). In comparing these 43 chloroplast genomes, they contained either 82 or 83 coding genes, 37 tRNA genes, and 8 rRNA genes, among which the genes *rpl2, rpl23, ycf2, ndhB, rps7,* and *rps12* have two copies due to duplication in the IR region. The exception to this is *D. oliveri* (MT644132) in which only a single copy of *ycf2* is found with the second copy having been truncated (Table 1; Supplemental Tables S1 and S2).

The *ndhF* gene spans the IRB SSC junction in most *Dalbergia* species (Supplemental Table S1). The SSC and IRA junction is spanned by *ycf1* in all 43 chloroplast genomes, while no gene spans the IRA and LSC junction. The intergenic region that spans the IRA and LSC junction varies widely in length across the 43 chloroplast genomes with the distance between *trnN-GUU* and *ndhF* ranging from 864 bp to 1668 bp. Even within a species, this region was found to vary as in *D. tonkinensis* with the distance ranging from 882 bp to 900 bp. Similarly, the intergenic region between *ycf1* and *trnN-GUU* varied from 411 bp to 429 bp across the *D. tonkinensis* chloroplast genomes (Figure 2, Supplemental Figure S2, Supplemental Table S1). The expansion and contraction of the SSC region across the samples used in this study might make this region a loci candidate for marker development.

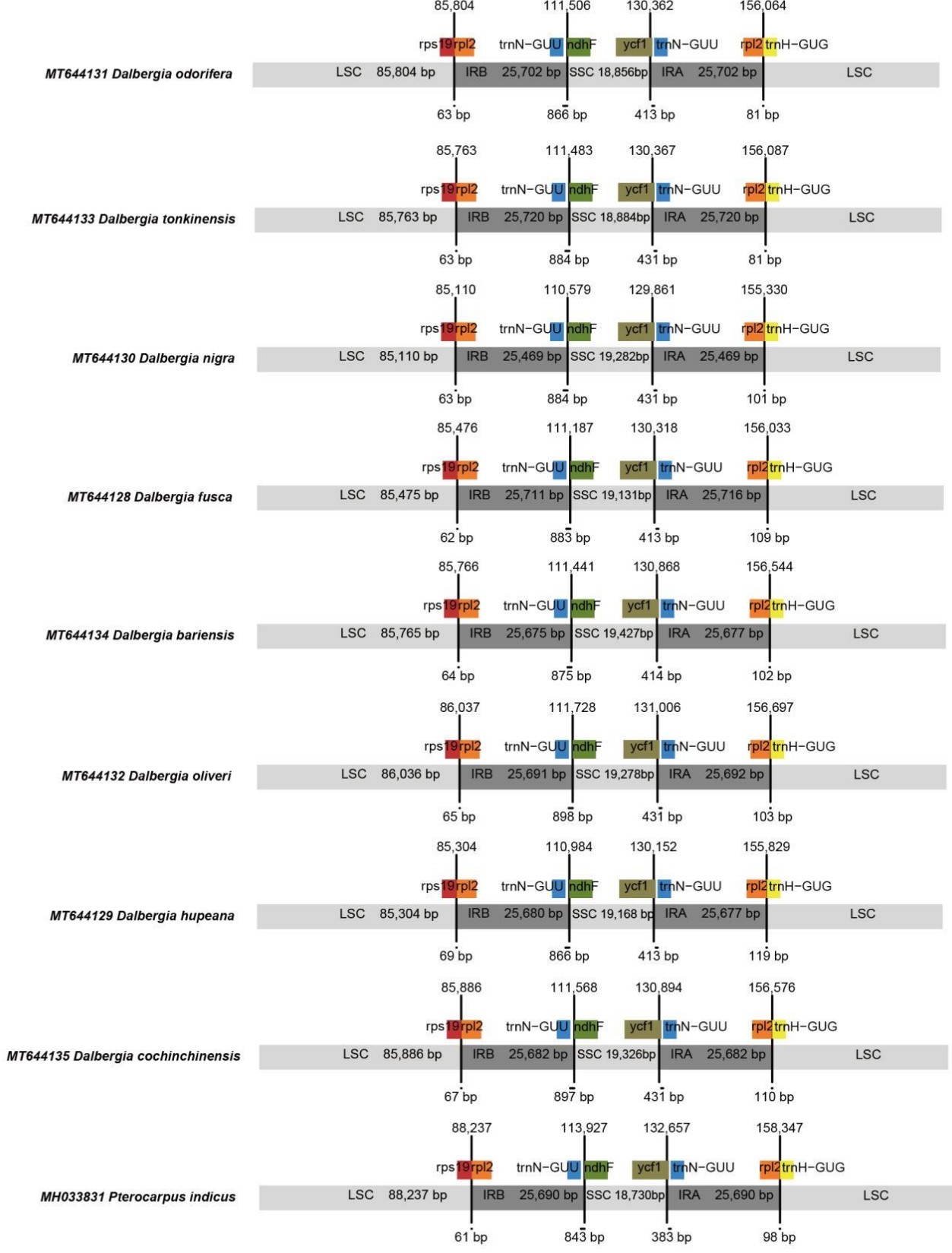

**Figure 2.** Comparison of junctions between the LSC, SSC, and IR regions among eight newly assembled *Dalbergia* species and the *Pterocarpus indicus* chloroplast genome. Figure is not to scale. (LSC: large single-copy, SSC: small single-copy, IR: inverted repeat). The numbers above the vertical line at the junction indicate the positions of the beginning and end of the IR region.

## 2.2. *Repeat Analysis*

Nucleotide repeats in chloroplast genomes can be very useful markers for identifying populations and/or species given the high rates of mutation in these regions. All 43 chloroplast genomes were analyzed using Reputer software using the limitation that repeats must have the length of 8 bp or more. Four types of repeats were considered, including forward (direct) (F), reverse (R), complement (C), and palindromic (P). Based on different motif types, the number of each type was counted based on grouping by sequence length (Figure 3). Almost all the repeats were in the 20–29 bp length range, followed by 30–39 bp, then 50+ bp, with the fewest in the 40–49 bp range. No C and R repeats were detected above 40 bp in length and they were rare even in the smaller size ranges. In the 30–39 bp group, C and R repeats were only found in a few species. The R repeats only appeared in six *Dalbergia* species, and C was only in *Pterocarpus*. For F and P repeat types, they were absent in the range of 40–49 bp in the scented rosewood species, while in the 20–29 bp group, the C type repeats were fewest in the Siam rosewood species (Figure 3). Given these differences in type and abundance, markers for identification of species and/or lineages could be devised based on different repeats. Fixed differences in repeat location, abundance, and type in a genome have provided ideal signatures for species or clade identification in previous studies [22,29,30].

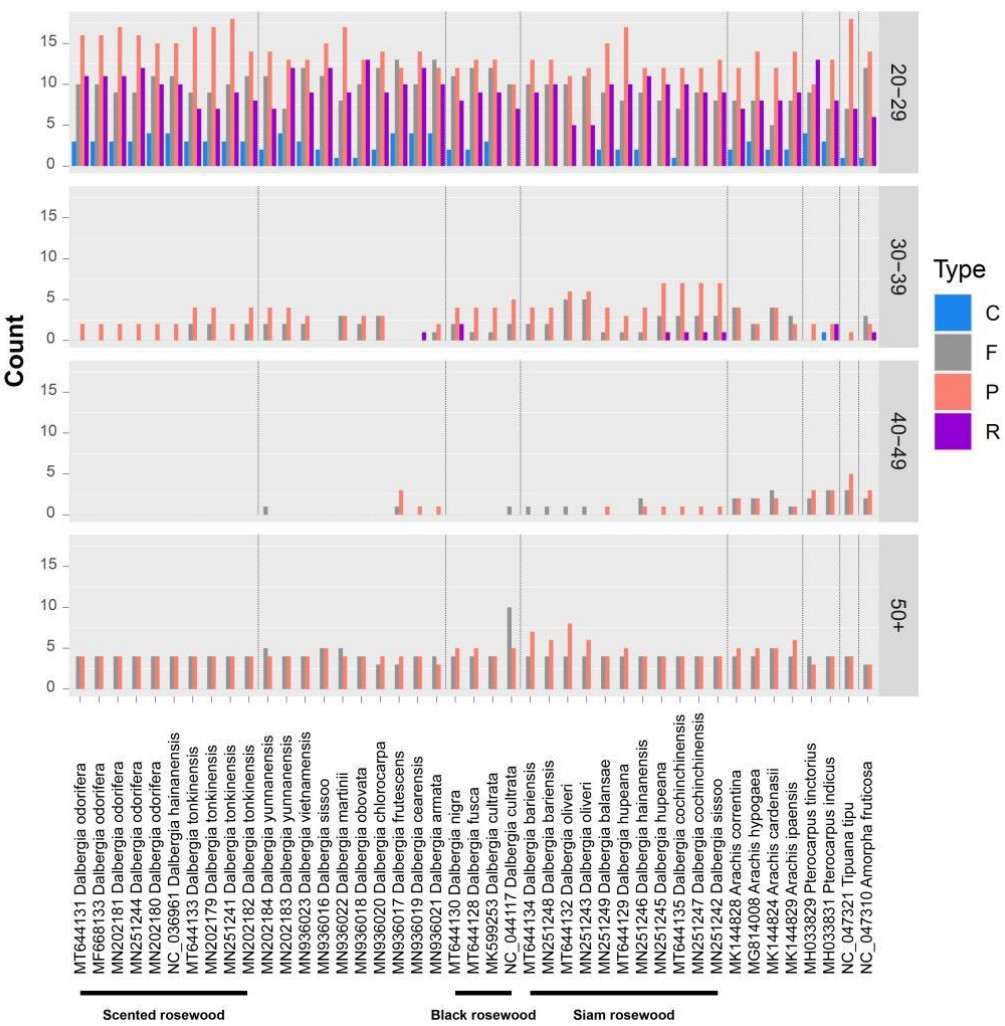

**Figure 3.** Variation in repeat abundance and type in 43 chloroplast genomes.

In chloroplasts, SSRs are often used for population genetics and/or phylogenetic analysis. Among all 43 chloroplast genomes, 86.7% (5579/6423) of the SSRs were single nucleotide A/T motifs. The genus *Arachis* was found to have less than half the number of A/T SSRs than the other species used in this study, suggesting that indels are more abundant in long A/T stretches for these species. Similarly, the numbers of nearly all other, save AG/CT and AGAT/ATCT, SSR motif types were far fewer in *Arachis* than the other species (Figure 4). Within *Dalbergia*, the number of SSRs varied greatly to the point where certain motifs such as AAT/ATT were present in the scented rosewoods and absent in the black rosewoods. Additionally, the presence or absence of certain SSRs differed within a given species such as AT/AT in *D. odorifera* and C/G in *D. tonkinensis*. The results from the SSR analyses suggests that these genomic regions might be useful for the identification of populations, species, and clades of *Dalbergia* if the data from different SSR motif types and length difference is combined in a nested analysis.

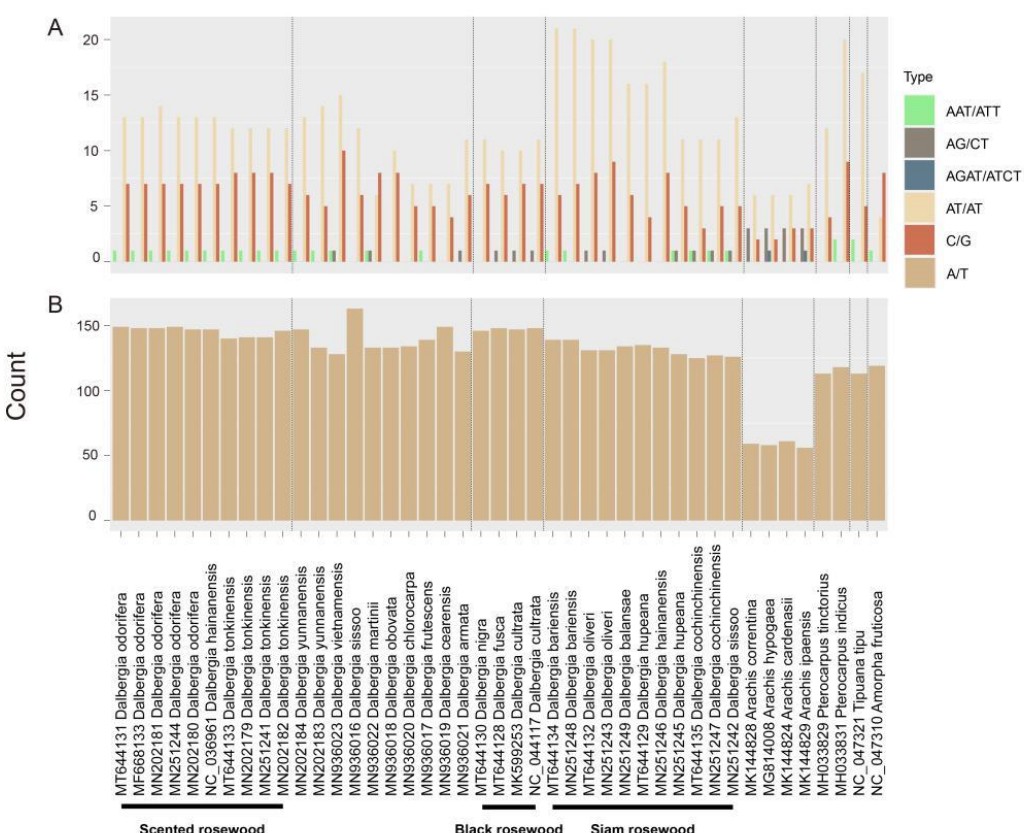

**Figure 4.** The number of simple sequence repeats (SSRs) of different types from 43 chloroplast genomes. (**A**) The number of AAT/ATT, AG/CT, AGAT/ATCT, AT/AT, C/G type SSRs; (**B**) The number of A/T type SSRs in each sample.

## 2.3. Genome Sequence Divergence

To further characterize the differences between chloroplast genomes, we employed mVISTA to find regions of greatest difference between conserved regions in the eight newly sequenced genomes and *Pterocarpus indicus* Willd. (an outgroup species that also produces high-quality rosewood-type lumber). The intergenic and intragenic regions were found to have the least similarity between chloroplast genomes in *Dalbergia* and *P. indicus* (Figure 5), especially in LSC (from *psbA* to *rps19*) and SSC regions (from *ndhF* to *ycf1*). Given these results, there are numerous intergenic and intragenic regions for developing markers to differentiate *Pterocarpus* from *Dalbergia* and fewer, but a sufficient number of regions to use within *Dalbergia* species for clade level differentiation.

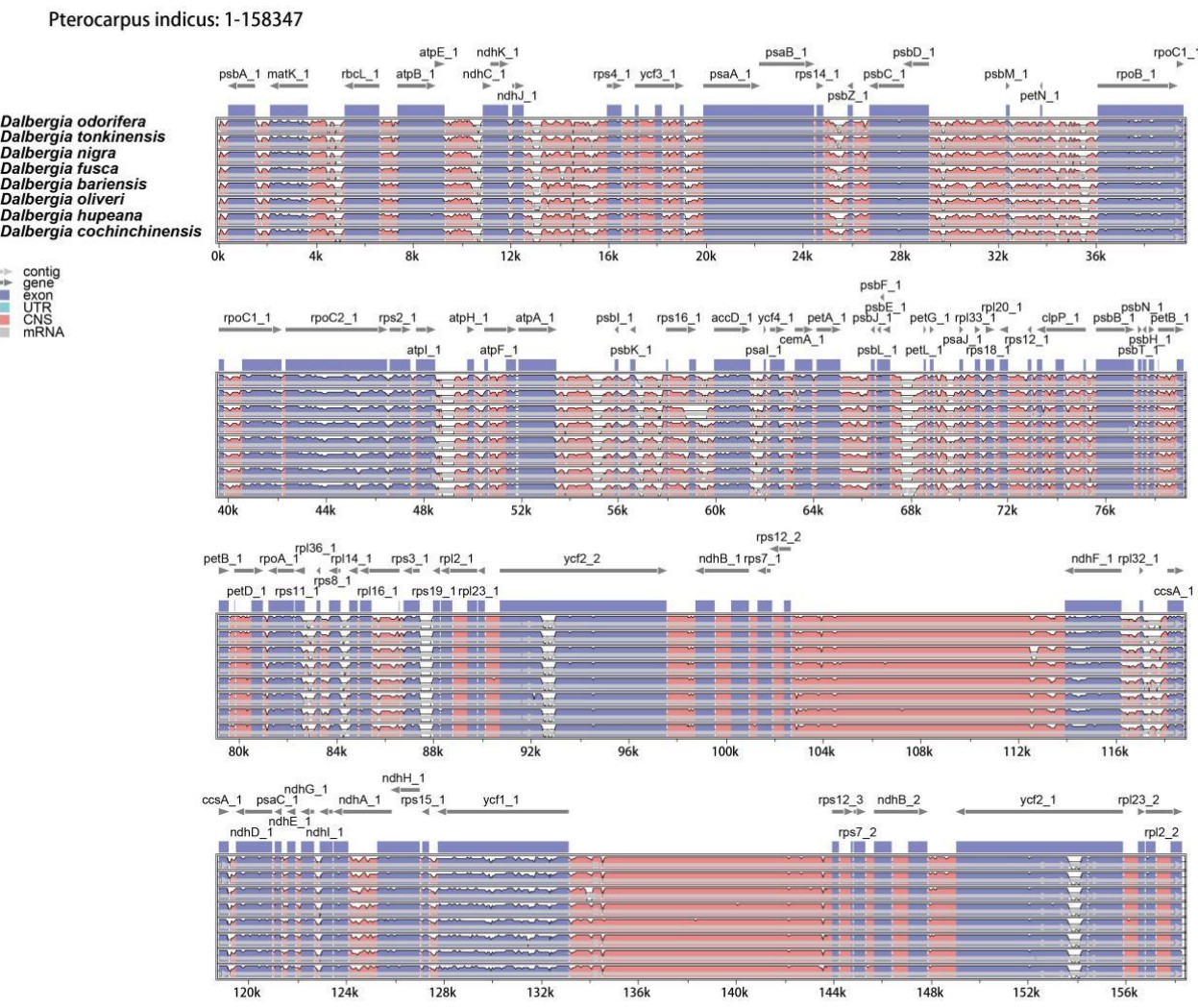

**Figure 5.** Global alignment of eight newly assembled *Dalbergia* chloroplast genomes using mVISTA with *P. indicus* as reference. *Y*-axis shows the range of sequence identity (50–100%). tRNA and rRNA genes were not present in this figure.

Based on the results of sliding window comparisons in Song (2019), we used T-Coffee to compare the complete genome sequences of all 43 samples in order to more comprehensively assess regions of dissimilarity (Figure 6, Supplemental Table S4). As in other plant lineages [31], CDS regions had very high identity scores across all 43 chloroplast genomes, while the lowest identity scores were found in *rps8-rpl14* (score 767; length 524 bp), *trnR-UCU-trnG-UCC* (score 788; length 592 bp), *accD-psaI* (score 790; length 824 bp), *psbA-trnK-UUU* (score 797; length 658 bp), and *ndhG-ndhI* (score 804; length 1308 bp). The identification of these hypervariable intergenic regions provide candidate regions for the development of genetic markers. While most intergenic regions are more variable, *psaA-psaB* (score 1000; length 25 bp), *psbL-psbF* (score, 1000; length, 22 bp), *psbF-psbE* (score, 1000; length, 9), and *ndhA-ndhH* (score, 1000; length, 1) were noted as being very similar and short in length. As such, these regions should be excluded from further consideration for marker development but might be useful in providing priming sites adjacent to variable regions in the development of molecular assays. Given that these photosynthesis genes are clustered in a single operon, strong selection of function has resulted in nucleotide conservation even in the intergenic regions. However, in general, sufficient nucleotide divergence has been found for the development of genetic markers in *Dalbergia* and to closely related genera that produce lumber of a similar appearance.

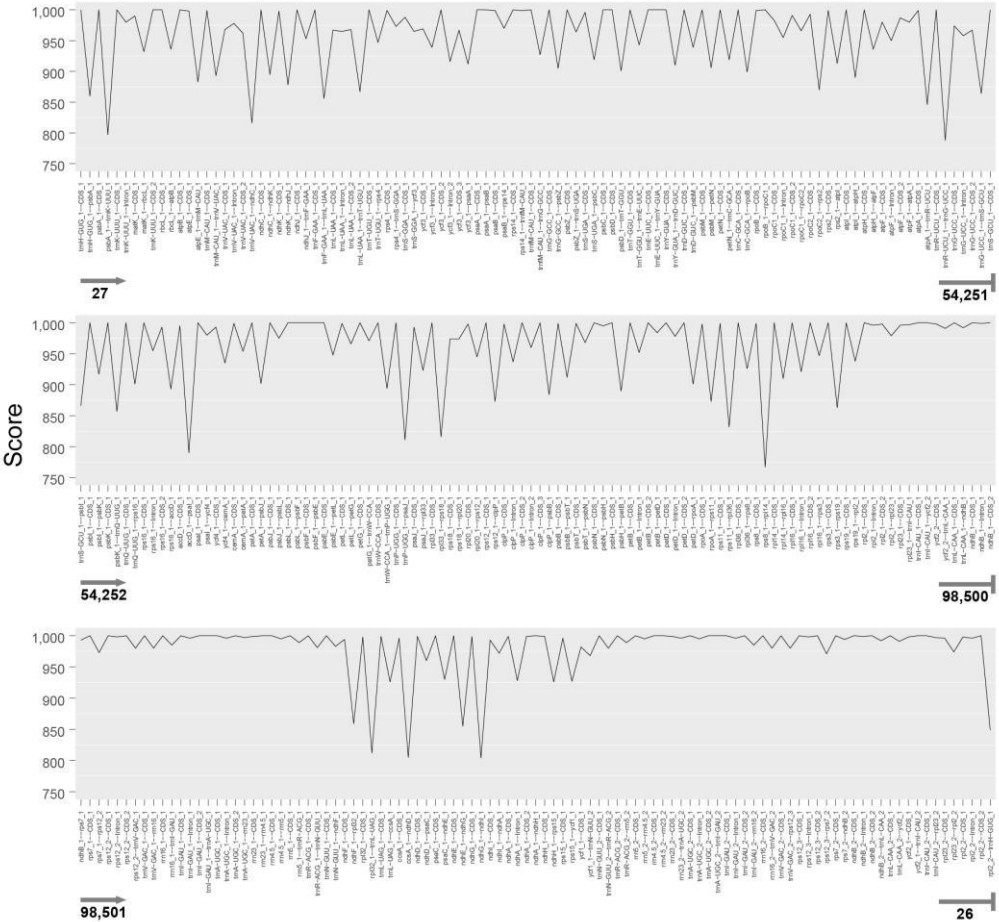

**Figure 6.** Sequence identity among coding and non-coding regions based on the alignment from 43 chloroplast genomes. T-Coffee was used to calculate the score of identity.

Based on the results of sliding window comparisons in Song (2019), we used T-Coffee to compare the complete genome sequences of all 43 samples in order to more comprehensively assess regions of dissimilarity (Figure 6, Supplemental Table S4). As in other plant lineages [31], CDS regions had very high identity scores across all 43 chloroplast genomes, while the lowest identity scores were found in *rps8-rpl14* (score 767; length 524 bp), *trnR-UCU-trnG-UCC* (score 788; length 592 bp), *accD-psaI* (score 790; length 824 bp), *psbA-trnK-UUU* (score 797; length 658 bp), and *ndhG-ndhI* (score 804; length 1308 bp). The identification of these hypervariable intergenic regions provide candidate regions for the development of genetic markers. While most intergenic regions are more variable, *psaA-psaB* (score 1000; length 25 bp), *psbL-psbF* (score, 1000; length, 22 bp), *psbF-psbE* (score, 1000; length, 9), and *ndhA-ndhH* (score, 1000; length, 1) were noted as being very similar and short in length. As such, these regions should be excluded from further consideration for marker development but might be useful in providing priming sites adjacent to variable regions in the development of molecular assays. Given that these photosynthesis genes are clustered in a single operon, strong selection of function has resulted in nucleotide conservation even in the intergenic regions. However, in general, sufficient nucleotide divergence has been found for the development of genetic markers in *Dalbergia* and to closely related genera that produce lumber of a similar appearance.

In a third approach to find regions of fixed differences for identifying *Dalbergia* species, we analyzed the dN (nonsynonymous substitution rates), dS (synonymous substitution rates), and the ratio of dN/dS (quantify strength of selection) of all genes with PAML. This approach is particularly useful in finding fixed differences that persist through a given lineage because mutations that are undergoing different modes of selection can be detected.

From this analysis, the dN of all genes was relatively low, while for the dS, two ribosomal genes (*rps16* and *rpl32*) were outliers. Except for *ycf2* (1.02), the dN/dS of all the other genes were less than 1, indicating that they are subject to purifying selection, especially photosynthesis genes, which were lower on average than the other four gene categories (Figure 7).

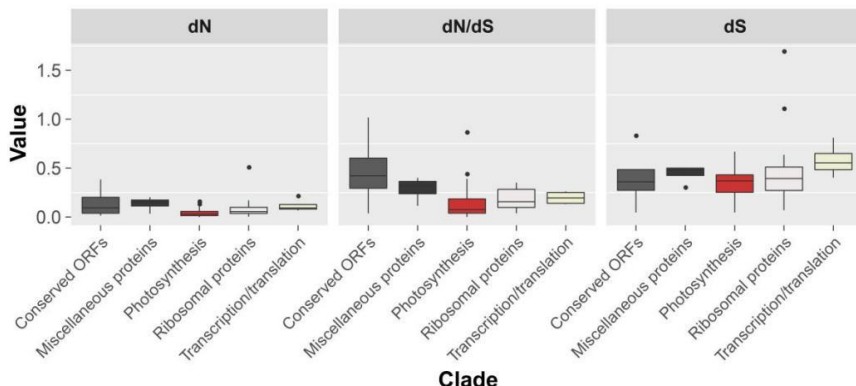

**Figure 7.** The mode and strength of selection among 77 chloroplast protein-coding genes in *Dalbergia*.

### 2.4. Codon Usage in Dalbergia Chloroplast Genes

Relative synonymous codon usage (RSCU) is often used to analyze the frequency of codon usage, with the higher values scaling with usage frequency (Figure 8). Given the conserved nature of codon usage, if a mutation is detected, it generally remains fixed and as such can provide a useful marker locus. Thus, we analyzed the codon usage of *Dalbergia* chloroplast genes. There were no large differences in RSCU between *Dalbergia* chloroplast genes, indicating conservation in codon usage across *Dalbergia*, and a very limited number of loci for clade or species identification.

### 2.5. Phylogenetic Analysis and Barcode Selection

In order to assess chloroplast genome divergence in an evolutionary context and find synapomorphies (and ultimately barcodes) for the given clades, we conducted a phylogenetic analysis using 77 homologous coding genes in 43 chloroplast genomes (Figure 9; Supplemental Figure S3). The resulting phylogenetic tree resolved several instances of polyphyly and paraphyly in regard to the species names applied to a given NCBI accession. For instance, *D. sissoo* MN936016 resolved in an early diverging position to a clade of *D. vietnamesis* P.H. Hô and Niyomdhan + *D. yunnanensis* Franch. + *D. tonkinensis* + *D. hainanensis* Merr. and Chun + *D. odorifera* with high support (BS = 100) and *D. sissoo* DC. MN251242 resolved in an early diverging position to a clade of *D. cochinchinensis* + *D. hupeana* with high support (BS = 100), making this species polyphyletic. Another clear example of polyphyly is *D. hainanensis* NC_036961 resolving in a clade with *D. odorifera*, while *D. hainanensis* MN251246 resolved in an early diverging position to a clade of *D. hupeana* + *D. balanense* Prain, making *D. hupeana* polyphyletic. Instances of paraphyly are also present, as in the branching order of *D. sissoo*, *D. cochinchinensis*, and *D. hupeana* in the Siam rosewood category, with other examples found elsewhere in the tree (Figure 9). These discrepancies between taxonomy and phylogeny indicate that some of the species identifications for *Dalbergia* NCBI accessions are probably incorrect. Alternatively, some of the examples of polyphyly might be the result of interspecific hybridization where a chloroplast genome was maternally inherited from a more distantly related species. Ultimately, in either case, improper identification (of hybrids and/or species) has led to discordance between the topology and taxonomic designations. These discrepancies should be addressed through identification of samples by taxonomic experts where possible as well as resequencing checks using nuclear loci (e.g., ITS) and reanalysis, as well as increasing the number of different species used in phylogenetic analyses. Furthermore, comparative phylogenetic approaches can be used to isolate which taxa are most likely misidentified in the chloroplast

data. For instance, *D. sissoo* MN936016 is probably correctly identified based on the location it resolved in the phylogenetic tree (early diverging to a clade containing *D. tonkinensis* in both phylogenies) in this paper compared to the position in the phylogenetic analyses of Vatanparast et al. [32] and Hassold et al. [20]. Issues of polyphyly in the Siam rosewood clade were further verified using whole chloroplast comparisons. Differences are apparent between the phylogenetic tree presented here and previously published phylogenies, as is expected given the different genomic regions and species sampled. That said, consistency is found among the two phylogenies in the membership of important taxa in separate clades and branching order. For example, *D. nigra* has an early diverging position to clade V in Vatanparast et al. [32] and to a clade with similar membership in the phylogenetic tree presented here. These similarities in phylogenetic topology suggest that both ITS and chloroplast barcoding could be employed to identify wood samples. However, much greater within-species sampling is needed to compare the stability of polymorphisms within species for each locus before a molecular assay can be deployed. This is especially true of ITS, where a large number of tandem copies can be found in multiple ribotypes [33] and potentially affect the accuracy of molecular assays. Lastly, it should be noted that in our analyses, the black rosewood category did not form a monophyletic grouping, whereas scented and Siam rosewood categories did form monophyletic groupings, provided the issue of incorrect labeling and polyphyly can be corrected.

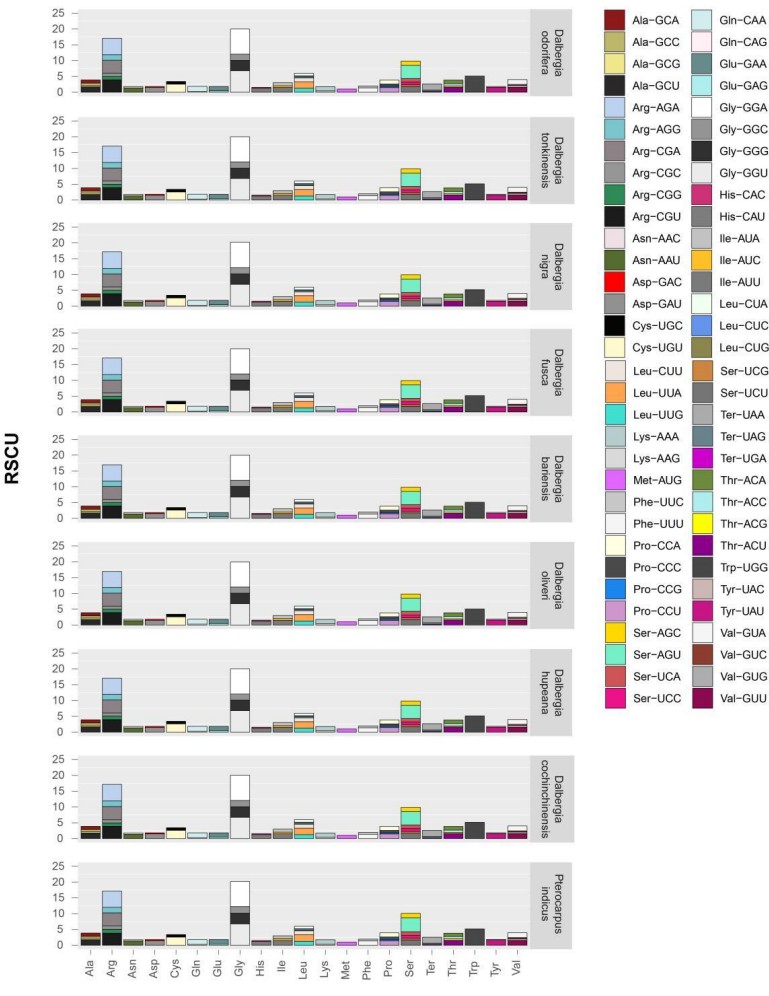

**Figure 8.** Codon content of 21 amino acids and a stop codon of 77 coding genes from 8 newly assembled *Dalbergia* chloroplast genomes. Color of the histogram corresponds to the color of codons in legend.

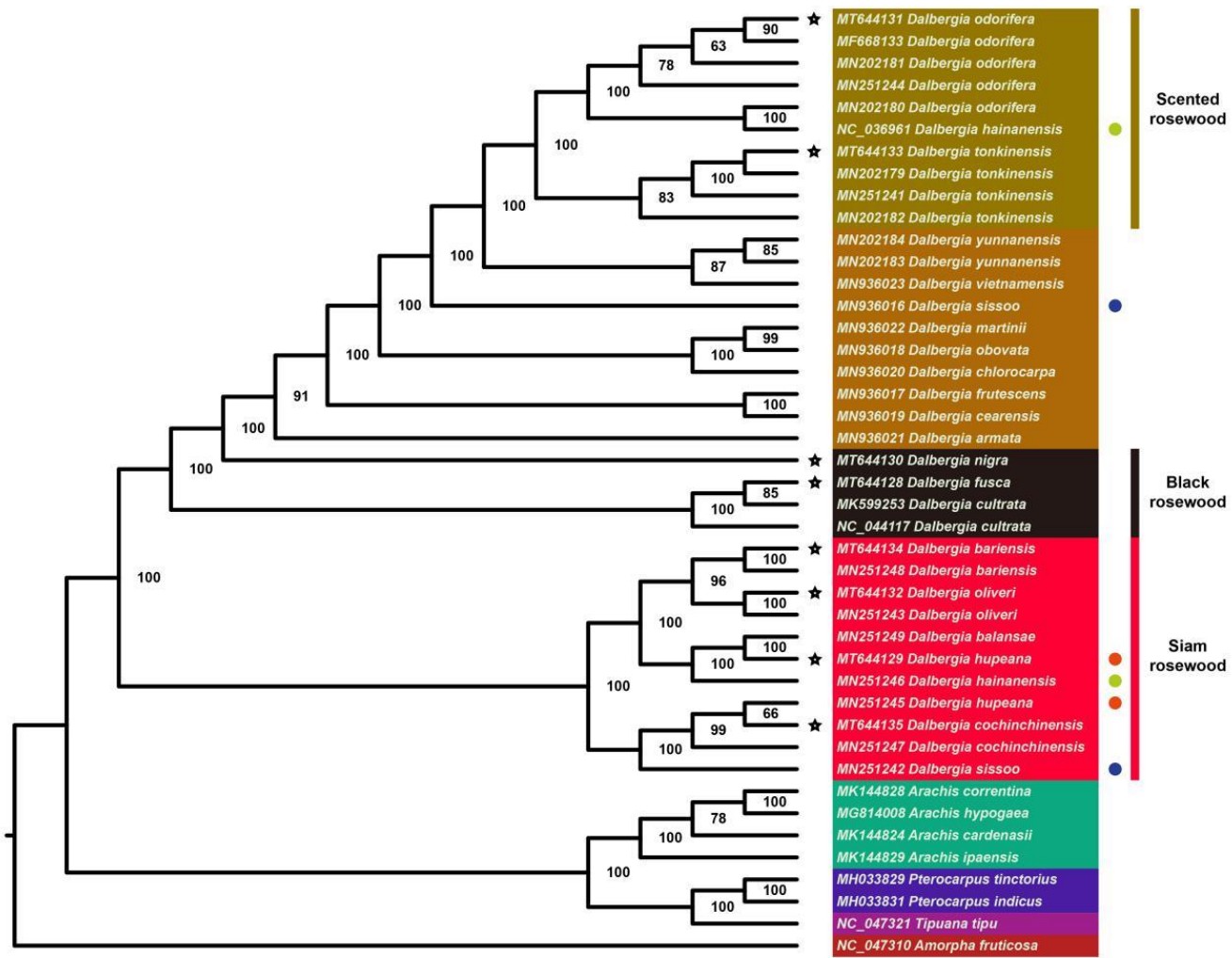

**Figure 9.** Phylogenetic tree of *Dalbergia* based on the CDS alignment of 77 chloroplast-coding genes. Numbers at nodes indicate the ultrafast bootstrap values generated by IQ-TREE. Species noted with stars are newly assembled chloroplast genomes from this study. Polyphyletic species are indicated with a dot.

From a MAFFT alignment of 43 chloroplast genomes, we scanned for loci rich in SNVs (single nucleotide variants) and INDELS (INsertions DELetions) to identify potential barcode loci (Supplemental Files S1 and S2). The MAFFT alignment after manual correction was 219,992 sites in total length. From this alignment, 58 SNV loci and 10 INDEL loci were found to identify Siam rosewoods, 29 SNVs and 13 INDELs in scented rosewood, and 2 SNVs and 1 INDEL for black rosewood if *D. nigra* was taken into account. If *D. nigra* was removed, 229 SNVs and 49 INDELs were found for the identification of the clade *D. cultrata* Benth. + *D. fusca*. However, since *D. nigra* is the highest priority *Dalbergia* species in CITES, a set of 94 SNVs and 69 INDELs were isolated for the identification of this species from all other *Dalbergia* used in this study (Table 2; Supplemental Tables S5 and S6). Given the arrayment of SNVs and INDELS across the entire genome, a super barcode approach could also be considered using the entire chloroplast genome to identify timber seizures from shotgun sequencing data [25]. As above, issues with polyphyly should be addressed to improve the confidence of these SNVs and INDELS in identifying a given *Dalbergia* subclade.

**Table 2.** Number of candidate molecular markers for black, Siam and scented rosewood.

| Group | Species Contained | Unique INDELs | Unique SNVs |
|---|---|---|---|
| Black rosewood | *D. fusca*, *D. cultrata*, *D. nigra* | 1 | 2 |
| Black rosewood | *D. fusca*, *D. cultrata* | 49 | 229 |
| Black rosewood | *D. nigra* | 69 | 94 |
| Siam rosewood | As shown in Figure 9 | 10 | 58 |
| Scented rosewood | *D. odorifera*, *D. tonkinensis*, *D. hainanensis* (NC_036961) | 13 | 29 |
| Scented rosewood | *D. odorifera* | 3 | 7 |
| Scented rosewood | *D. tonkinensis* | 3 | 0 |

## 3. Materials and Methods

### 3.1. Tissue Samples and DNA Extraction

We collected fresh leaves from eight species of *Dalbergia* for DNA extraction. The leaf material from plants were collected at the Experimental Station of the Research Institute of Tropical Forestry, Chinese Academy of Forestry, Jianfeng Town, Ledong Li Autonomous County, China. Detailed source and preservation information for these species is listed in Supplemental Table S7. To obtain the chloroplast genome sequences, the genomic DNA was extracted by QIAGEN DNeasy Plant Maxi Kit (Cat. NO 68163) for Illumina paired-end sequencing.

### 3.2. Genome Sequencing and Assembly

The Illumina HiSeq 2500 platform was used to sequence the extracted DNA with insert sizes of 500 bp, for 150 bp paired-end read lengths. Raw data was quality control filtered using Trimmomatic [34] with the following criteria: filtered reads with adapters, filtered reads with N bases >10%, and filtered reads with low-quality bases ($\leq$5) >50%, which yielded 2 Gb of clean reads for each species.

All paired-end clean reads were aligned to the chloroplast database (containing all published chloroplast genomes from NCBI by the date 26 November 2019) with bwa v0.7.17-r1188 [35] software, and then the Picard v2.20.3 program was used to select chloroplast reads. The selected chloroplast reads were assembled by Spades v3.14.0 [36] with default parameters and the output scaffolds (GFA file) were imported into Bandage v0.8.1 [37] to generate the final chloroplast genome for each species.

### 3.3. Genome Annotation

All 43 chloroplast genomes, which contained 8 newly assembled *Dalbergia* chloroplast genomes, and 27 other published genomes of Dalbergieae, 4 *Arachis* L. species, 2 *Pterocarpus*, 1 *Tipuana* Benth. species, and an outgroup species *Amorpha fruticosa* L., were compiled for analysis, with all NCBI accession numbers listed in Supplemental Table S1. The *D. tonkinensis* genome was used as a reference, with all genes delimited manually to provide a complete reference template. The re-annotation of all species was then executed using Plastid Genome Annotator (PGA) [38], and the visualization of genome structure was implemented by the Draw Organelle Genome Maps online software (OGDRAW) [39].

### 3.4. Genome Structure Analysis

Four repeat types in 43 chloroplast genomes, F (forward), P (palindrome), R (reverse), and C (complement) were identified using REPuter [40] with default settings. Simple sequence repeats (SSRs) were detected using the Perl script MISA [41], with 10, 6, 5, 5, 5, and 5 repeat units set for mono-, di-, tri-, tetra-, penta-, and hexa-motif microsatellites set as the minimum threshold, respectively. CodonW v1.4.4 [42] was employed to assess codon distribution on the basis of relative synonymous codon usage (RSCU) ratio.

### 3.5. Genome Nucleotide Diversity

Analyses of genome sequence diversity was performed using an online software mVISTA [43] to compare the 8 newly assembled *Dalbergia* species using Shuffle-LAGAN [44] alignment program, with the *P. indicus* chloroplast genome used as a reference. All 43 chloroplast genomes were split into several parts based on annotation files, and the overall consistency score of each part was calculated with multiple sequence alignment tools using T-Coffee [45] in default mode.

### 3.6. Phylogenetic Analysis and Nucleotide Substitutions

The whole chloroplast genome sequence alignment of 43 chloroplast genomes was generated using MAFFT v7.464 [46,47] software, with TrimAL v1.4 [48] used to trim the poorly aligned positions. The longest CDS sequences of 77 protein-coding genes were extracted from each genome according to the annotation files and also aligned using MAFFT [46,47]. The nucleotide sequence alignments of 77 protein-coding genes were concatenated. This data set was further used to resolve the phylogenetic tree using IQ-TREE v2.0 [49,50] with 1000 ultrafast bootstrap replicates to assess branch support with FigTree v1.4.3 (http://tree.bio.ed.ac.uk/software/figtree, accessed on 15 June 2020) used for tree visualization. CODEML in PAML v4.9 [51] was used to calculate the nonsynonymous (dN), synonymous (dS), and the ratio of nonsynonymous to synonymous nucleotide substitutions (dN/dS) for each gene.

**Supplementary Materials:** The following supporting information can be downloaded at: https://www.mdpi.com/article/10.3390/f13040626/s1, Figure S1. Chloroplast genome structure of 8 newly assembled *Dalbergia* species. Figure S2. Comparison of junctions between the LSC, SSC, and IR regions among *Dalbergia odorifera* and *D. tonkinensis* chloroplast genome. Figure S3. Phylogenetic tree of *Dalbergia* based on the CDS alignments of 77 chloroplast genes with branch lengths. File S1. Final alignment file for 43 chloroplasts used in this study. File S2. The Perl scripts used to identify potential barcode loci. Table S1. Genome information of the 43 chloroplast genomes used in this study. Table S2. The N ratio for 43 chloroplast genomes used in this study. Table S3. The gene number of chloroplast genomes from *Dalbergia*. Table S4. Sequence identity of the region from 43 chloroplast genomes that has been split up. Table S5. The SNP sites and the nucleotide change in different rosewood clades. Table S6. The INDEL sites and nucleotide change in different rosewood clades. Table S7. Detailed source and preservation information for species.

**Author Contributions:** Z.H. conceived and designed the study. W.H., Z.W., and X.L. (Xuezhu Liao) performed the experiments and data analysis. X.L. (Xiaojing Liu) contributed materials. Z.W. and X.L. (Xuezhu Liao) wrote the paper. L.R.T., Z.W., X.L. (Xuezhu Liao), and D.X. revised the paper. All authors have read and agreed to the published version of the manuscript.

**Funding:** This study was co-supported by the Research Funds for the Central Non-profit Research Institution of Chinese Academy of Forestry (CAFYBB2020SZ005), National Natural Science Foundation of China (31500537), and the Chinese Academy of Agricultural Sciences Elite Youth Program to Zhiqiang Wu (JCKY2020-07).

**Institutional Review Board Statement:** Not applicable.

**Informed Consent Statement:** Not applicable.

**Data Availability Statement:** The chloroplast genome sequences are available in the NCBI database, under accession number MT644128–MT644135.

**Acknowledgments:** We sincerely thank Shanghai BIOZERON Biotechnology Co., Ltd. for performing the high-throughput sequencing.

**Conflicts of Interest:** The authors declare no conflict of interest.

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
