# Peer review of "Comparative Analyses of 35 Complete Chloroplast Genomes from the Genus Dalbergia (Fabaceae) and the Identification of DNA Barcodes for Tracking Illegal Logging and Counterfeit Rosewood"

_forests, doi:10.3390/f13040626_

Round 1

Reviewer 1 Report

General comments

This paper presents the results of sequencing chloroplast genomes from 8 species of rosewood, a genus involved in illegal logging.  Overall it is a well-written manuscript that fits well with the scope of the journal.

The manuscript could be improved by reformatting the introduction. Most of the species sequenced in this study have already had their chloroplasts sequenced and published. The new sequences presented here are still useful but the introduction should be re-written to reflect what this particular study adds to the research field i.e., which species are newly sequenced here and which are additional chloroplast genomes of already sequenced species that will be useful for testing species monophyly. If the herbarium vouchers for these specimens can be included (see comment below) then this could also be included in the intro and emphasised as it distinguishes this study from Song et al (2019). Furthermore, the Hassold et al (2016) paper on this genus that is cited later in the manuscript should also be cited in the introduction to frame this new work. Their study had a similar aim but only sequenced three chloroplast loci, which were unable to identify species. Their results justify sequencing whole chloroplasts to obtain better resolution in this genus.

The biggest issue I have is with the availability of data. Firstly, and most easily to solve, is that the raw sequencing reads should be deposited in a public repository, such as GenBank’s SRA, and an accession number provided in the manuscript.

Potentially more difficult to achieve is that herbarium vouchers of the 8 sequenced specimens need to be deposited in a registered herbarium. This is especially important given some of the sequences are quite different to published chloroplast sequences purported to be from the same species. These new sequences are effectively meaningless if they cannot be linked to a morphological voucher whose identification can be verified. Note that the previous study of chloroplasts from this genus  (Song et al 2019) also did not record vouchers and the present study is just perpetuating the problem.

Specific comments

Ln 69: change to: …based on a lack of morphological differences in confiscated...

Ln 74: AFLP requires a large amount of good quality DNA, which is unlikely to be obtained from wood. You could consider incorporating this information or delete mention of AFLP.

Ln 87: another advantage in this context that could be mentioned is that the high copy number of chloroplasts means chloroplast DNA is more likely to be detected from wood products, which likely have low levels of DNA that is degraded – see https://pubmed.ncbi.nlm.nih.gov/19414167/

Ln 118: Add ‘chloroplast’ before genome sequences.

Table 1: the columns LSC, IRB, SS and IRA are presumably in base pairs (bp)? This info should be added to the title of the table or to the column headings.

Ln 147: delete ‘of’ before Dalbergia

Ln 222: change ‘where’ to ‘were’

Ln 249: change ‘general’ to ‘generally’

Ln 250: change ‘Given this’ to ‘Thus’

Ln 253: add ‘suitable’ after ‘loci’

Ln 286: change ‘trees’ to ‘phylogenies’

References

There is inconsistency with the how journal titles are cited – sometimes each word in the title is in capital letters but other times in lowercase.

Ln 425: It is usual to record the date accessed for online databases

Lns 430, 434, 437 and 451: genus and species names should be in italics.

The references are cut off part way through the Nurk et al reference.

Author Response

Response to Reviewer 1 Comments

Response: Thanks for all constructive comments and corrections, we corrected some mistakes in the writing and adjusted the unclear sentences to make the article easier for readers to understand.

This paper presents the results of sequencing chloroplast genomes from 8 species of rosewood, a genus involved in illegal logging. Overall it is a well-written manuscript that fits well with the scope of the journal.

Point 1: The manuscript could be improved by reformatting the introduction. Most of the species sequenced in this study have already had their chloroplasts sequenced and published. The new sequences presented here are still useful but the introduction should be re-written to reflect what this particular study adds to the research field i.e., which species are newly sequenced here and which are additional chloroplast genomes of already sequenced species that will be useful for testing species monophyly. If the herbarium vouchers for these specimens can be included (see comment below) then this could also be included in the intro and emphasised as it distinguishes this study from Song et al (2019). Furthermore, the Hassold et al (2016) paper on this genus that is cited later in the manuscript should also be cited in the introduction to frame this new work. Their study had a similar aim but only sequenced three chloroplast loci, which were unable to identify species. Their results justify sequencing whole chloroplasts to obtain better resolution in this genus.

Response 1: Thank you for your suggestions. We have sequenced 8 chloroplast genomes of Dalbergia, among which Dalbergia Nigra from black rosewoods has never been reported, while the other 7 are additional chloroplast genomes of sequenced species. This has been well demonstrated in Figure 9. Meanwhile, we also added some descriptions in introduction to make the statement clearer. Here, we mainly focused on black rosewood while Song et al. (2019) mainly sequenced the species of scented rosewood and siam rosewood, and we selected more species of Dalbergia to make evolutionary relationships representative. Hassold et al. (2016) mainly analyzed partial sequences for DNA barcoding (matK, rbcL, trnL (UAA)). Although the results could play a good role, it also had limitations. For example, the success of species identification based on single barcode marker or combination is low, while whole chloroplast genome was studied in our research, which will provide more information. Besides, we also added this part to the corresponding position of Introduction.

Point 2: The biggest issue I have is with the availability of data. Firstly, and most easily to solve, is that the raw sequencing reads should be deposited in a public repository, such as GenBank’s SRA, and an accession number provided in the manuscript. Potentially more difficult to achieve is that herbarium vouchers of the 8 sequenced specimens need to be deposited in a registered herbarium. This is especially important given some of the sequences are quite different to published chloroplast sequences purported to be from the same species. These new sequences are effectively meaningless if they cannot be linked to a morphological voucher whose identification can be verified. Note that the previous study of chloroplasts from this genus  (Song et al 2019) also did not record vouchers and the present study is just perpetuating the problem.

Response 2: We have uploaded the whole chloroplast genomes and corresponding annotations information to the NCBI database and provided the accession numbers in this manuscript. We believe that the information needs to be compared are well displayed on the assembled chloroplast genome. In terms of raw data, the nuclear genome assembly of these species is being performed, and we haven't released the raw data yet. Based on the reviews from the published chloroplast genome paper, they all submitted the assembled chloroplast genome. We also followed the same rules. The raw short-sequenced reads may supply more genome information, we are currently assembling the whole genome of those species. We will supply all those data to NCBI in near fugure.

Point 3: Specific comments

Ln 69: change to: …based on a lack of morphological differences in confiscated...

Response 3.1: We corrected this.

Ln 74: AFLP requires a large amount of good quality DNA, which is unlikely to be obtained from wood. You could consider incorporating this information or delete mention of AFLP.

Response 3.2: Thanks for your suggestions, we have deleted it.

Ln 87: another advantage in this context that could be mentioned is that the high copy number of chloroplasts means chloroplast DNA is more likely to be detected from wood products, which likely have low levels of DNA that is degraded – see https://pubmed.ncbi.nlm.nih.gov/19414167/

Response 3.3: Thanks for your constructive suggestion, we have added it.

Ln 118: Add ‘chloroplast’ before genome sequences.

Response 3.4: Thank you, we added it.

Table 1: the columns LSC, IRB, SS and IRA are presumably in base pairs (bp)? This info should be added to the title of the table or to the column headings.

Response 3.5: Thanks the location of these region has been noted in Figure 2, and we added the legend in Figure 2 to make the content clearer.

Ln 147: delete ‘of’ before Dalbergia

Response 3.6: Corrected.

Ln 222: change ‘where’ to ‘were’

Response 3.7: Corrected.

Ln 249: change ‘general’ to ‘generally’

Response 3.8: Corrected.

Ln 250: change ‘Given this’ to ‘Thus’

Response 3.9: Corrected.

Ln 253: add ‘suitable’ after ‘loci’

Response 3.10: Corrected.

Ln 286: change ‘trees’ to ‘phylogenies’

Response 3.11: Corrected.

There is inconsistency with the how journal titles are cited – sometimes each word in the title is in capital letters but other times in lowercase.

Response 3.12: Corrected.

Ln 425: It is usual to record the date accessed for online databases

Response 3.13: Thank you for your advice, but maybe the location you described does not correspond with the location in my original submission. But if you mean the time when we downloaded the data of the published chloroplast genomes, here we have recorded the accession numbers for these published chloroplast data in Supplemental Table 1. The detail of each recording chloroplast genome could be check online NCBI database.

Lns 430, 434, 437 and 451: genus and species names should be in italics.

Response 3.14: Corrected.

The references are cut off part way through the Nurk et al reference.

Response 3.15: We have added it again, maybe there is a problem with the system conversion, our original submitted version contains all the references.

Reviewer 2 Report

The manuscript presents 8 new plastid assemblies of Dalbergia and makes a comparative analysis to other previously published sequences from genbank. The nature of the data is quite incremental without any big scientific breaktrough, nevertheless the experimental part, analysis and presentation is very well performed, as well as the interpretation of the results and completeness. The manuscript is well-organised and there are no or very few language problems. 

I belive the paper can be published with very few editing and minor changes only at editorial level. 

Author Response

Response to Reviewer 2 Comments

The manuscript presents 8 new plastid assemblies of Dalbergia and makes a comparative analysis to other previously published sequences from genbank. The nature of the data is quite incremental without any big scientific breaktrough, nevertheless the experimental part, analysis and presentation is very well performed, as well as the interpretation of the results and completeness. The manuscript is well-organised and there are no or very few language problems. 

I belive the paper can be published with very few editing and minor changes only at editorial level. 

Response: Thank you for your comments, and we reread the article and corrected all mistakes.

Round 2

Reviewer 1 Report

The authors have addressed most of my queries and I can now see the entire reference list. However, herbarium vouchers are still not provided for their new sequences, which is important given the confusion over species identification in this group.

Note also the instructions for authors for Forests states:

 For research manuscripts involving rare and non-model plants (other than, e.g., Arabidopsis thaliana, Nicotiana benthamiana, Oryza sativa, or many other typical model plants), voucher specimens must be deposited in an accessible herbarium or museum. Vouchers may be requested for review by future investigators to verify the identity of the material used in the study (especially if taxonomic rearrangements occur in the future). They should include details of the populations sampled on the site of collection (GPS coordinates), date of collection, and document the part(s) used in the study where appropriate.

Author Response

Response to Reviewer 1 Comments

Response: Thanks for all constructive comments, we added the detailed sampling information.

The authors have addressed most of my queries and I can now see the entire reference list. However, herbarium vouchers are still not provided for their new sequences, which is important given the confusion over species identification in this group.

Note also the instructions for authors for Forests states:

For research manuscripts involving rare and non-model plants (other than, e.g., Arabidopsis thaliana, Nicotiana benthamiana, Oryza sativa, or many other typical model plants), voucher specimens must be deposited in an accessible herbarium or museum. Vouchers may be requested for review by future investigators to verify the identity of the material used in the study (especially if taxonomic rearrangements occur in the future). They should include details of the populations sampled on the site of collection (GPS coordinates), date of collection, and document the part(s) used in the study where appropriate.

Response 1: Thank you for your suggestions. We added Supplemental Table 7 containing detailed information about the sample sources, planting years and preservation sites.
